# The Role Personal Responsibility Norms Play in Sustainable Development for University Students: The Impact of Service-Learning Projects

**Susana Lucas Mangas** [1,*]**, José María Marbán** [2] **, María Concepción Unanue Cuesta** [3]**,**
**María Ángeles Manso Argüelles** [1] **and José Romay Martínez** [4]

[1]    Department of Psychology, University of Valladolid, 47011 Valladolid, Spain; jaomsae@hotmail.com
[2]    Department of Didactics of Experimental Sciences, Social Sciences and Mathematics, University of Valladolid, 47011 Valladolid, Spain; josemaria.marban@uva.es
[3]    Department of Sociology and Social Work, University of Valladolid, 47011 Valladolid, Spain; mariaconcepcion.unanue@uva.es
[4]    Department of Psychology, University of A Coruña, 15071 A Coruña, Spain; jose.romay@udc.es
[*]    Correspondence: susana.lucas@uva.es

**Abstract:** Sustainable development is a desirable, global challenge that is as complex as it is necessary. While individual actions are positive, effective progress must start from a collective commitment. In line with this, recent research points to an urgent need for an increased effort to make sustainability education a key element in the basic literacy of all people. This study takes on this challenge of environmental education and changing attitudes in order to reach sustainable community development while relying on the principles of the value-belief-norm model (VBN). A quasi-experimental research design was used to analyze the impact of learning service projects on the activation of personal norms of university students. The results show that although no statistically significant differences appear between pre-test and post-test measurements, there is a clear trend towards improvement of personal norms about sustainable development, which encourages further research in this area.

**Keywords:** attitudes; environment; service-learning; norm; personal responsibility; higher education; SDG learning; sustainability competences

## 1. Introduction

On the 25th of September 2015, the General Assembly of the United Nations adopted the Education 2030 Agenda for Sustainable Development, a "plan of action for people, planet, and prosperity" meant to uphold universal peace and access to justice [1].

Spain, along with 192 other member states of the United Nations, passed a resolution that highlighted the elimination of poverty as the current and most important world challenge, affirming that without this, sustainable development cannot advance.

The Education 2030 Agenda, an agreement that includes 17 Sustainable Development Goals (SDGs) and 169 integral and indivisible goals, serves as both a guide for the elaboration of service-learning projects as well as diverse university dynamics that contribute to sustainable development.

Sustainable development is rooted in the eradication of poverty, the fight against inequalities across countries as well as the conservation of the planet and the creation of continued, sustainable, inclusive economic development, which is tightly linked to and symbiotic with social inclusion (A/RES/70/1 United Nations: Transform Our World: Education 2030 Agenda for Sustainable Development).

Along the same lines, several of UNESCO's World Declarations on Education [2–5], emphasize that society is faced with the challenge of promoting quality education rooted in social responsibility while establishing cooperative dialogue between universities and society aimed at the facilitation of sustainable community development and the Culture of

Peace, all through the role of mediation to help advance service-learning projects aimed at sustainable human–community development [6]. The concept of Culture of Peace was born in Yamoussoukro, Africa, in 1989 at a UNESCO congress. It is closely connected to well-being, harmonious relationships for progressive respect, and also to nature, which is, irrevocably summed up in a word: health. Sustainable development is considered a basic human right [6] and is recognized in Article 28 of the Universal Declaration of Human Rights of 1948. It continues to be a core issue of various reports and declarations from international organizations such as: UNESCO [2–5]; 2030 Agenda [1]; UNDP [7,8]; the university guide to the incorporation of SDGs [9]; Official Journal of the European Union [10]; and the CEDEFOP [11].

In this way, and under the slogan "My Education, My Right(s)", the Global Campaign for Education (GCE) emphasized the role education can play in the fight against environmental deterioration and work in favor of a model for sustainable development. The GCE is made up of more than 124 countries from around the world. It is an international coalition of diverse organizations all active in the fight to conserve the agreements signed by different states which guarantee all people universal access to quality education. In the case of Spain, GCE consists of 54 organizations and, since its inception, includes annual participation in the Global Action Week for Education (GAWE, in Spain: SAME), which aims at increasing societal awareness of human rights and the elaboration of educational materials worked on throughout the year. Given that each SDG is specified and is considered reached as long as human rights are respected, a clear connection is established between the SDGs and education as a basic human right.

The research project presented in this article, strongly linked to the central theme of the GCE-2019 (CME-2019, Spain education for environmental and social sustainability), has been made possible thanks to the collaboration of the Office of Environmental Quality and Sustainability of the University of Valladolid (UVa) and the support of an agreement between the Ministry of Development and Environment, the local government of Castilla y Leon and the UVa. This agreement sustains the funding granted to information and environmental education programs linked to environmental management, such as those associations who promote environmental literacy in the aforementioned university and have taken the lead in the Teaching Innovation Project "Legal Clinic, Service-learning for the Protection of Human Rights: Culture of Peace: Education for Environmental and Social Sustainability" (Legal Clinic and Observatory of Human Rights at UVa) in conjunction with the International Cooperation Area for University Development.

In line with all of the above, the main objective of this study is to analyze to what ex-tent service-learning projects impact on some standards of personal responsibility in university students according to the value-belief-norm (VBN) model and in connection with sustainable development. The VBN is a generalization of the norm activation model (NAM) by Schwartz [12] to explain altruistic and environmentally friendly behavior. More specifically, considering the fact that education is a key issue for sustainability, participants were selected from the Degree in Primary Education and the Degree in Social Education at two different campuses at UVa. Although both studies are concerned with the training of professionals in the educational field, the profile of both degrees varies in terms of competences and in terms of the contexts in which graduates develop their actions. Therefore, special attention to potential differences between the impact of service-learning projects within both groups is paid.

Thus, regarding the main aim of this research project as stated above, two research questions arise: To what extent does in-class implementation of service-learning projects linked to social and environmental sustainability in university education impact on the personal responsibility norms of students in relation to sustainability? Is there any difference between the way such service-learning projects impact students enrolled in educational degrees with different professional profiles, more specifically in primary pre-service teachers and in social educators in initial training?

From such research questions, the following statistical hypothesis were stated to be tested:

**Hypothesis 1 (H1).** *No significant differences exist between the measurement of personal responsibility norms of participants before and after the intervention.*

**Hypothesis 1 (H2).** *No significant differences exist between the averages of the norms of personal responsibility of students enrolled in the Degree in Primary Education or those enrolled in the Degree in Social Education before and after the intervention.*

To end this section, it is worth mentioning that service-learning projects considered for this research were based on the general objectives of the Global Campaign for Education, 2019 [13] (http://www.cme-espana.org) (accessed on 24 January 2019), upholding the same competencies, objectives, and specific contents of various subject areas in which the previously mentioned service-learning projects were developed. In particular:

1. To show the different environmental and social challenges that humanity currently faces, the interconnection between them and how they affect the well-being and the exercise of the people's rights around the world.
2. To improve knowledge about the role that education plays in empowering citizens to jointly build a socially and environmentally sustainable future for all people.
3. To promote a change in people's attitudes and behaviors in order to advance in the construction of a sustainable future from a social and environmental perspective.

Correspondingly, the following Sustainable Development Goals (SDGs), established as priorities in the GCE-2019, guided the choice and design of the service-learning projects to be considered for experimental intervention: quality education (SDG4); sustainable cities and communities (SDG11); responsible production and consumption (SDG12); climate action (SDG13); underwater life (SDG14); life of terrestrial ecosystems (SDG15); and partnerships to achieve the goals (SDG17).

## 2. Theoretical Basis

Different reports on criteria for the inclusion of issues related to sustainability in university study plans [14,15] maintain that it is necessary for universities to favor responsible social and civil practices that combine academic learning with community service, which should be in accordance with the betterment of quality of life and social inclusion. Being a stakeholder in service-learning—a methodology that boosts academic development and competency skills—means taking on service-learning for development of student's competencies related to social responsibility [16] and in accordance with the revision of good practices of service-learning in the field of intervention compiled from several networks and references such as Red Española de Aprendizaje-Servicio and the Red Universitaria de Aprendizaje-Servicio, the European Observatory of Self-Learning in International Higher Education, the Association for Research on Service-Learning and Community Engagement (IARSLCE), CLAYSS, and many research contributions [17–21].

*Service-Learning*

Service-learning "is a social intervention strategy through education (within training projects or research) which articulates the learning and service process together with the community (whether it be geographically close or far, on a local level or a more global spectrum, or even world-wide), in an innovative project on cooperative dialogue, in which the participants experience getting involved in the needs and challenges of their environment with the goal of contributing to sustainable human–community development. This promotes quality of life and helps link both students (as they take on the "learners role as a mediator") and the educational organizations involved (in this case, the university) with society, all this through mutual learning in which both parties help and are helped, building an interdependent relationship through planning, critical reflection, and shared evaluation of relevant, complete information. This becomes more and more systematic as the project unfolds, in distinct organizational phases (preparation, planning, execution, evaluation-follow up)" [6].

On the other hand, as some authors [22] point out, human rights have an interesting place between social problems and aspirations of well-being and quality of life; they are not only a question of laws, politics, and large control structures. They are most likely and fundamentally a question of attitudes, social images that are shared or not, values, socialization processes, expectations, and goals, to only name a few deep psychosocial aspects.

In this same vein, various research studies consider the need for an increased effort to help make sustainability education influential on attitudes and personal responsibility norms especially those of university students, and moreover, future educators [23–25].

In this context, the current study springboards from previous studies on sustainable behavior explained through values, beliefs, personal norms, and how they directly influence behavior [22,24,26–29]. This study works from the theoretical bases that guides Psychology of Environmental Education as presented in the New Human Interdependence Paradigm (NHIP) and branches from the concept of interdependent development that supports sustainability in the environmental balance.

Contributions from different sources [30–32] were taken into consideration such as the Theory of Reasoned Action (TRA) and the Theory of Planned Behavior (TPB), which argues that people act according to the information and knowledge they have on a certain subject, which, in this case, is the environment. As part of the intervention accompanying this research project, all students were given the information necessary on the environment and social and environmental sustainability.

According to the VBN model, values have an influence on the development of a person's general beliefs about the environment and how to relate to nature. This affects specific beliefs about the environment, causing a higher or lower level of awareness of the consequences of one's actions on the environment (AC) and one's ascription of responsibility (AR), finally leading to the activation of one's feeling of moral obligation to the environment, called personal norm (PM) [33].

Durán, Alzate, and Sabucedo [17], on the other hand, believed Chu and Chiu [32] to be the first authors to defend the influence norms and personal obligations have on the variable perception of importance and moral obligation, in this case, related to the separation of rubbish. This variable "reflects the perception one has of whether or not separating the rubbish is ethically correct or incorrect and, besides, reflects the interiorized pressure consistent with their own value system" [32] (pp. 604–626). Without moral obligation, which in turn leads us to act responsibly, it will be very difficult or even impossible to stop environmental degradation of our planet, as the studies previously mentioned underline as well.

## 3. Methods

Based on the aforementioned information, it is clear that the impact of service-learning projects on personal responsibility norms of university students related to their orientation to sustainability and their pro-environment behavior greatly depends on the in-class and out-of-class activities they experience and the way in which education intervention carries them out.

In that way, it is important to point out that the preparation and planning phases of the intervention associated with this research project were essential and were developed beforehand in the first semester of the 2018–2019 academic year; in fact, six months of planning went into these phases and included pre-design tasks and pilot activities, for example, documentary analysis of educational resources offered by GCE-2019 [13]. At the same time, considering the importance of the role of the teacher in the development process of this project, great coordination and training efforts were made in these initial phases which, in turn, ensured a high level of teacher cooperation with the implementation of the action while minimizing any disturbance of the internal validity that could occur due to individual differences among teachers.

The intervention itself was carried out and completed during the second semester and consisted of the elaboration of tutored projects on environmental sustainability in small research cooperative groups of 4–5 students.

With all this in mind, different didactic proposals were prepared in order to work on students' diverse transversal competencies, all in accordance with the Global Campaign for Education 2019. All competencies linked up with the theoretical content learned in the subjects involved.

The main learning outcomes expected to be achieved by students participating in this project were linked to the development of competences related to problem solving, linguistics, social and civic, cognitive, anticipatory and strategic skills, cooperative learning, critical thinking, personal independence and initiative, learning to think and communicate, relating to others, maintaining personal balance, and creating. Other various areas outlined in the Sustainable Development Goals (SDG) in the Education 2030 Agenda were included with the help of references from didactic resources of the GCE-2019: "Defend Education, Hold up the World," and more specifically, those in the central theme called "Social and Environmental Sustainability in University Education."

The community service project, organized around didactic units and panels of experiences recorded in diaries, reports, and portfolios, consisted of a combination of case studies, presentations, debates, images, stories, songs and the elaboration of texts and videos of theatre plays and other art, student reports, workshops, guided readings, conceptual maps, posters, graphs, diagrams, workshops, self-questionnaires, and cooperative projects, all of which contributed to visualizing thoughts, feelings–emotions, and actions. These educational resources were made possible with the cooperation of the Entreculturas Foundation and with all the organizations that form part of the Global Action Week For Education (GAWE; in Spain, SAME: https://somos.entreculturas.org/general/semana-de-accion-mundial-por-la-educacion-2019-en-valladolid) (accessed on 30 January 2019), which took place at the Faculty of Education and Social Work and included cooperative workshops together with the community, social entities, and educational centers.

Finally, the evaluative process of each action included teachers and students as participants as well as the Citizenship Regional Technical staff of Castilla y Leon at the Entreculturas Foundation, taking into consideration all the project dimensions and aiding in the reflection of all the learning results achieved about the social impact of service and courses of action. For that, criteria of mixed evaluation was established:
  - Well-defined objectives
  - Predicted/reached indicators regarding project objectives
  - Global assessment of activities and expected results
  - Level of impact on educational resources created in the community area
  - Verbal, written, and expository clarity of the students' projects
  - Clarity and quality of resource materials used in the presentations

Learning outcomes and the impact of service on real contexts were evaluated through a combination of heteroevaluation, co-evaluation, and self-evaluation procedures by means of a variety of evaluation tools which included portfolios, essays, reflective diaries, panels of experiences, systematic observation checklists, written tests on theoretical and practical concepts, questionnaires, and rubrics. Furthermore, evaluation was a joint task between teachers, students, and the Citizenship Regional Technical staff of Castilla y Leon at the Entreculturas Foundation. This was done in order to determine whether they followed all the learning and community service action criteria whose impact was subject to evaluation; conducted in such a way that the differences would not affect, in the first place, the potential impact on the beliefs of the students participating in the study. Actions and timelines for the research project are summarized in Figure 1.

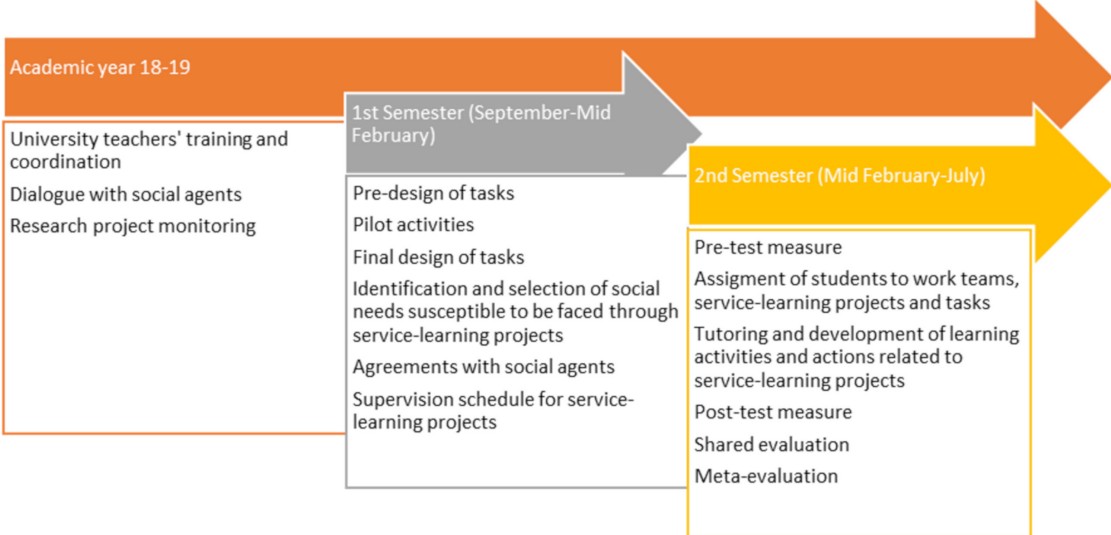

**Figure 1.** Actions and timelines for the research project.

*3.1. Sample*

The selection of students participating in this study was carried out by means of a convenience sampling so that those responsible for teaching such students were also those who had participated in the preparation–planning phases described in the previous section. In total, there were 210 participants, 156 (74%) were students doing their Degree in Primary Education and 54 students (26%) were enrolled in the Degree in Social Education. More specifically, 65 were 1st year students, 84 were in their 2nd year, and 61 were 3rd year students.

*3.2. Instrument*

In order to collect the data necessary and contrast the proposed hypothesis, a scale developed by Steg, Deeijerink, and Abrahamse [33], and taken from Hernández, Ruiz y Suárez [34], was used. This instrument was suitable for the measurement of personal responsibility norms to be evaluated within the VBN model and was adjusted to a classic Likert scale format with 7 responses (ranging from "totally disagree" to "totally agree") and 9 items (Table 1) that center on the three well-known components that make up the measurement of N (norm), which can be one dimensional or broken into: personal norm, awareness of the consequences, and specific beliefs. In this study, the Cronbach Alpha for the scale was 0.868, thus showing a high level of internal consistency.

**Table 1.** Scale items.

| Items | Component |
|---|---|
| 1   Today's lifestyle causes people to consume too much | Specific beliefs |
| 2   I can contribute to reducing the wastefulness of today's society by reducing the things I buy or reusing them | Awareness of consequences |
| 3   I would feel guilty if I bought things when I didn't really need them | Personal standard |
| 4   I am concerned about the current level of consumption in society | Specific beliefs |
| 5   I feel morally obliged to reduce what I buy and reuse what I already have | Personal standard |
| 6   I can help to reduce current consumerism by avoiding waste | Awareness of consequences |
| 7   By reducing the things I buy and reusing them I can contribute to reducing the problems of consumerism | Awareness of consequences |
| 8   The level of production and consumption in today's society is a serious problem | Specific beliefs |
| 9   I feel I am doing the right thing when I reduce my purchases and reuse what I have | Personal standard |

*3.3. Procedure*

A quantitative approach by means of a quasi-experimental design without a control group was chosen in order to be able to respond to the proposed research questions, taking into account the conditions that the project had to assume, specifically resources and time available. In this way, all the participating students were given a pre-test containing the Likert scale described above. Neither teachers nor students were given the results of the first data collection in order to avoid unwanted influence of knowing how others would act. When the intervention phase was finished, the same survey was given to the participating students in the post-test phase.

## 4. Results

This section will show the results obtained in terms of impact on the personal responsibility norms (within the VBN model) of students after the service-learning project were carried out, as described previously, analyzing this with data from the application of the scale before the intervention and after the intervention.

Before showing the results of the scale, it is important to note that the number of participants in the post-test (126 from primary education and 37 from social education) was smaller than the one for the pre-test (156 from Primary Education and 54 from Social Education), showing a slightly higher experimental death ratio in one of the degrees (see Figure 2), which is normal in studies such as this one. Concerning the year of study, the experimental death ratios were 24.6%, 31%, and 9% for 1st, 2nd, and 3rd years, respectively.

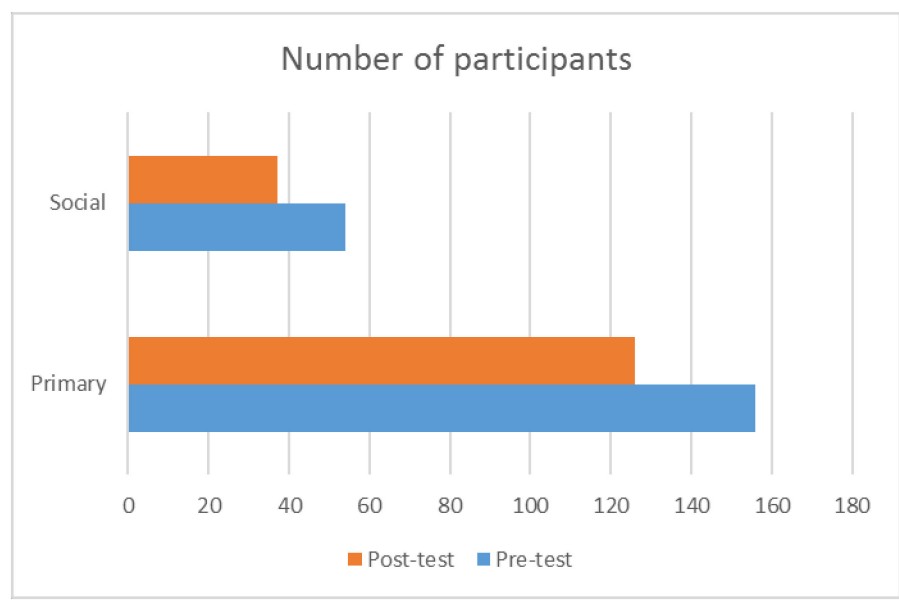

**Figure 2.** Participants according to degree in pre-test and post-test phases.

A contrast of measurements based on t-tests for the relevant samples was used in order to analyze the results. Each participating student was given a code to identify themself. The level of significance was fixed to 95%. In Table 2, one can see the statistics of groups according to the moment of data collection (pre-test and post-test) together with the results coming from the t-test for mean differences, while Table 3 collects results associated to the degree-intragroups mean comparisons as was established in H2. In all cases, the Levene test was applied to ensure equality of variances.

**Table 2.** Group statistics and results from *t*-test.

| Time of Data Collection | N | Mean | SD | SE for Mean | 95% Confidence Interval for Mean | t-Value | Sig. (Bilateral) |
|---|---|---|---|---|---|---|---|
| Pre-test | 210 | 41.88 | 7.26 | 0.50 | (40.89,42.87) | −1.47 | 0.14 |
| Post-test | 163 | 43.04 | 7.78 | 0.61 | (41.83,44.24) | | |

**Table 3.** Results of the *t*-test for degree-intragroups' mean comparisons.

| Group | t | Sig. (bilateral) | Mean Difference Pre-Test/Post-Test | 95% Confidence Interval for the Difference | |
|---|---|---|---|---|---|
| Primary | −1.00 | 0.31 | −0.89 | −2.66 | 0.86 |
| Social | −1.68 | 0.09 | −2.44 | −5.33 | 0.44 |

One can observe that, although the average value obtained in the post-test was higher than that of the pre-test phase, this difference is not statistically significant ($p > 0.05$), meaning the null hypothesis cannot be discarded.

In this case, we observe that none of the subgroups of students according to the degree they were enrolled in changed their mean values in a significant way; therefore, H02 cannot be rejected either.

However, an analysis by course level did reveal significant differences between the pre-test and post-test values in the case of the 2nd year students of the Degree in Social Education (Table 4), who represent 38 participants in the pre-test data collection and 18 in the post-test.

**Table 4.** Results of the contrast of averages in second year students.

| t | Sig. (bilateral) | Mean Difference | 95% Confidence Interval for the Difference | |
|---|---|---|---|---|
| −2.36 | 0.02 | −1.16 | −9.34 | −0.76 |

Through a more detailed analysis of the responses (Figure 3) one can observe, on one hand, a positive evolution of students' personal norms, centering on the measurement scale produced in each and every one of the items making up the scale, with exception of the first, with differences in mean that range between 0.15 (E6) and 0.29 (E3). On the other hand, the biggest change was observed in the student responses corresponding to E2, E3, E5, and E9, items that integrate the sub-dimension associated with personal norms and thus collecting affirmations associated with sense of responsibility and personal moral.

The results obtained, even if they do not statistically confirm the initial hypothesis in all the courses, do in fact show tendencies that seem to point in that direction. Likewise, the previously mentioned learning outcomes and the service to the community offered together with the CME-2019 [13], indicate that it is worth speculating in this direction, with the aim to develop the norm of personal responsibility.

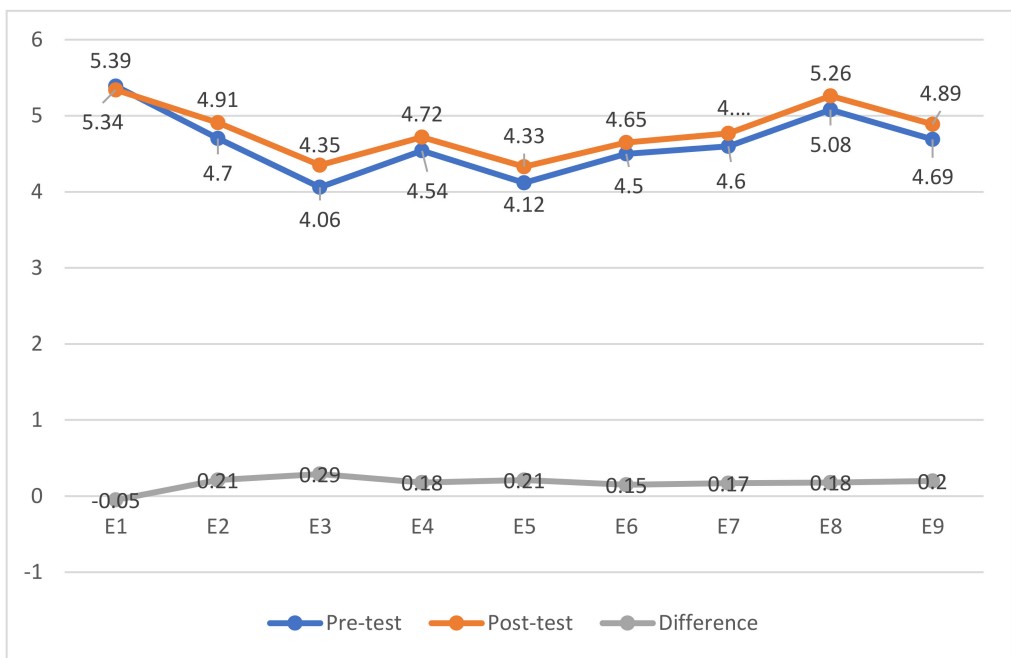

**Figure 3.** Average values of each item on the scale.

## 5. Discussion and Conclusions

An analysis of the results of this study points to the tendency to develop personal responsibility as described by the VBN model through environmental education [22,23,25,34,35] and the development of service-learning projects [16,17,21,36,37]. Significant differences were only revealed in one of the participant groups of this study.

Reflecting on the differences between these groups, and more specifically, on the significant difference found in the aforementioned group which was made up of second year students pursuing a Degree in Social Education, their more developed sense of responsibility (according to the VBN model) could be explained by their leadership in the Meeting for the Global Action Week for Education (GAWE/SAME-2019: https://somos.entreculturas. org/general/semana-de-accion-mundial-por-la-educacion-2019-en-valladolid) (accessed on 30 January 2019), which took place in the Department of Education and Social Work at the University of Valladolid. All the participating groups in this study attended this event, but only the 2nd year Social Education Degree students took on the role of university student representatives. The event consisted of cooperative workshops, which had been previously planned together with social entities, and was attended by private users, students from educational centers, and municipal councillors, all of whom participated in the reading of the GAWE/SAME-2019 manifesto.

These results reinforce previous results about how to develop prosocial behavior from service-learning approaches [18].

The social impact of this community service event reinforced the norms of personal responsibility at this point in their education in favor of social and environmental sustainability [12] and responded to the related recommendations by UNESCO [2–5], the European Observatory for Service-learning [38], and CRUE [14] for universities to commit to the improvement of quality of life in the community. This challenge is met through the creation of service-learning projects [13], that develop research focus, establish alliances to contribute to the achievement of the SDGs linked to education in favor of environmental and social sustainability, and are in line with the 2030 Agenda [1,8].

Thus, we stand before a new conception of social and environmental sustainability, which has been deeply accepted in recent years and has been a model for some of the studies referenced here as well as the model for the scale used in this research. Behind it, there is a scaffolding of values, beliefs and attitudes, which comes from this type of com-munity

service-learning actions with the community and time, and being done continuously through time, thus helping to internalize this responsibility norm to a greater extent.

The positive trend shown by the results obtained here points to an adequate approach in terms of methodology and educational strategy, which opens up new possibilities for intervention. This research has intertwined a series of actions that seek to develop personal responsibility within the field of education for sustainability. Future studies will be able to deepen the analysis of the impact of each specific action to achieve said educational and social aims.

The 2030 Agenda gained strength beginning with the United Nations Conference on Human Environment in 1972 up to The United Nations Summit on Sustainable Development in 2015, and is the culmination of more than four decades of dialogue, and multilateral debate on how to respond to environmental, social, and economic challenges on the international community level.

The 2030 Agenda is based on the fundamental principles of: universality, leave no one behind, interconnection and indivisibility, inclusion and cooperation. At the same time, there are five fundamental dimensions to the agenda: people, prosperity, planet, collective participation, and peace.

These five pillars are directly rooted in the fact that, in order for a development project to be sustainable, it should consider any social, economic, and environmental repercussions it could cause as well as support conscious decision making with respect to pros and cons, synergies, and any possible side-effects it could produce (all in agreement with the objectives and competencies referred to in the main part of the GCE/CME-2019 [13]).

At the same time, those responsible for the creation of policy should assure that all interventions in development are managed and carried out with the pertinent alliances and mobilized with the necessary resources for its execution, just like this study proposed through this cross-disciplinary service-learning project which promotes: analysis, cooperation, critical thinking, and participatory, creative social innovation that encourages the exploration of different and innovative approaches, and the redefinition of fundamental aspects of the Sustainable Development Goals.

Although the hypothesis of this study was not fully confirmed, one can see that after the intervention an important change occurred in all items that measure personal responsibility norms. This opens up interesting future lines of research in this area like that of Durán, Alzate, and Sabucedo [17] in which the personal responsibility norm is a fundamental variable with regards to positive change in attitudes and behaviors related to environmental sustainability. Social research not only has certain limitations to the internal validity but also to the external validity due to the fact that results found are discrete and not easily generalizable. Upon analysis, the following limits to this particular study have been found: high experimental mortality or loss of experimental subjects, which could indicate that students who participated in the post-test phase were those with higher interest. This potentially could have influenced the results in favor of the impact of educational intervention on the personal responsibility norm in the VBN model. Nevertheless, it is important to say that experimental mortality did not affect students in different groups yet mainly affected one group that, in general and for unexpected reasons, was unable to participate in the post-test data collection. This limited the final results of the study by fundamentally reducing its size and the range of representation in the sample. However, it did not limit in terms of possible bias in the profile of the student participants who completed and did not complete the post-test.

To that effect, it is important to support initiatives that show signs of transforming personal responsibility norms (according to the VBN model) in education and, through their cross-disciplinary character, focus on evaluating the results of the service on social impact and quality of learning. These projects, at the same time, should foster better collaboration and participation, which is outlined in the Sustainable Development Goals for Social and Human Sciences. Furthermore, these projects should generate spaces for knowledge exchange and meetings, necessary tools in the creation of an autonomous, empowered

society based on the ability to understand and take on sustainable development and the Culture of Peace, which are two very important aspects that should be integrated into in-class work [21,36,37,39,40].

According to Lucas [6], service-learning training projects seek to respond to sustainable human–community development, which is not only a methodology, a teaching-learning technique and a pedagogical approach, but also an innovative teacher and researcher strategy for psychosocial intervention and community development. Students' learning must relate to socially and community responsible service carried out in a co-operative way. The university thus becomes a social agent, one that is co-responsible for the environment.

The use of this entrepreneurial social innovation proposal is being promoted at the Spanish university level [41,42], and is part of the identity of the University of Valladolid.

The Conference of Rectors of Spanish Universities, CRUE [14], developed a document that values social responsibility and service-learning as an ideal teaching methodology for the development of skills in university sustainability and social responsibility. Specifically, the University of Valladolid, where this project is registered, has been seeking intra-university coordination of different actions that work on social responsibility since the incorporation of service-learning in the Strategic Plan 2008–2014 (4th Axis: the university in society). Currently, there is the creation and reactivation of the Delegation of the Rector for university social responsibility. The existence of this stable structure of support for the projects makes it easier to give continuity to them over time. In addition, this provides added value to the training of educators, who in the future will intervene professionally in different socio-educational environments and have a greater impact on all citizens.

**Author Contributions:** Conceptualization, S.L.M., M.C.U.C. and J.R.M.; methodology, J.M.M.; validation, S.L.M., M.C.U.C. and J.M.M.; formal analysis, S.L.M., J.M.M., M.C.U.C., M.Á.M.A. and J.R.M.; investigation, S.L.M., J.M.M., M.C.U.C. and M.Á.M.A.; resources, S.L.M.; data curation, J.M.M.; writing—original draft preparation, S.L.M., J.M.M., M.C.U.C., M.Á.M.A. and J.R.M.; writing—review and editing, S.L.M., J.M.M. and M.C.U.C.; supervision, S.L.M., M.C.U.C. and J.M.M.; project administration, S.L.M.; funding acquisition, S.L.M. All authors have read and agreed to the published version of the manuscript.

**Funding:** This research was funded by the Office for Environmental Quality and Sustainability of the University of Valladolid under the agreement signed between the Ministry of Development and Environment of Junta de Castilla y León and the University of Valladolid which supports information and environmental education programs linked to environmental management as well as actions that promote environmental literacy at the university.

**Institutional Review Board Statement:** The study was conducted according to the guidelines of the Declaration of Helsinki and the University of Valladolid ethical standards for research.

**Informed Consent Statement:** Informed consent was obtained from all subjects involved in the study.

**Data Availability Statement:** The data presented in this study are available on request from the corresponding author.

**Acknowledgments:** The authors acknowledge all the support given to this project by students and social entities that made possible the development of the service-learning projects used as educational intervention.

**Conflicts of Interest:** The authors declare no conflict of interest.

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
