# Peer review of "The Role Personal Responsibility Norms Play in Sustainable Development for University Students: The Impact of Service-Learning Projects"

_sustainability, doi:10.3390/su13137330_

Round 1
Reviewer 1 Report
This manuscript summarizes an intervention to discover how to develop standards of personal responsibility according to the VBN model in students
through service–learning projects based on the general objectives of the Global Campaign for Education 2019. It is well contextualized in reference to relevant sources and models of Education for Sustainability. However, only a few references comply with the journal's rules for citation and references.
In the material and methods it would be necessary to identify more clearly the features of the intervention, and link evaluation crteria with specific evaluation tools. Figure 1 is not explicit enough about the structure of the proposal.
As for the research questions, why is the reason why there should be a difference among grades? I.e., why does it make sense to check H0.2? MOreover, the justification should come from the introduction.
The sample would better be described as "convenience" (the students that were accesible to the researchers were used).
Although the scale is based on a published source, it would be convenient to report some psychometric indices.
Likewise, the (meagre) results could be presented in a much more efficient way: i.e., group means (SD), student's t value, 95% range and p at once, for each of the comparisons. Otherwise, it gives the misleading impression taht results are much richer than they in fact are. And, in contrast, only a sentence (lines 295 - 296) are used to clear up the second H0.
In the end, none of the differences are statistically significant, so there is little evidence of the effectivity of the approach. In the discussion it is said that only one of the groups showed differences, when it is only one course of one of the degrees. As such, the effect is truly of small size.
For increasing the interest of the paper, it is suggested that the intervention, including the structure, scope and arrangement of the service learning projects is better described, and maybe some evidence of the allegued learning objectives provided.
Both English and general punctuation should be revised throughout for enhanced readability.
Author Response
Dear Reviewer:
Below you will find a list of the changes we have introduced in our original manuscript Sustainability-1225211 entitled The influence of personal responsibility norms on changes in attitudes towards the environment through a service-learning project as an answer to the review comments (we have highlight all the changes made in the manuscript by using the track changes mode in MS Word). We appreciate all the contributions made in the review process considering all of them useful and interesting, having allowed their reading and analysis an improvement, in our opinion, in the final version of the manuscript.
Estimado revisor:
A continuación encontrará una lista de los cambios que hemos introducido en nuestro manuscrito original Sustainability-1225211 titulado La influencia de las normas de responsabilidad personal en los cambios en las actitudes hacia el medio ambiente a través de un proyecto de aprendizaje-servicio como respuesta a los comentarios de la revisión (hemos destacado todos los cambios realizados en el manuscrito mediante el modo de seguimiento de cambios en MS Word). Agradecemos todas las aportaciones realizadas en el proceso de revisión considerándolas todas útiles e interesantes, habiendo permitido su lectura y análisis una mejora, en nuestra opinión, en la versión final del manuscrito.

Reviewer 2 Report
The paper addresses an importnat but thorny topic - how to ensure education is effective in addressing our sustainability challenges. The stated aim of this paper is how to (L85) develop standards of personal responsibility according to the VBN model in students through service–learning projects. Using the VBN model itself is an interesting approach. However the focus of the paper is a descriptive quantitative analysis of the results of service learning, in terms of changes in their measured personal responsibility norms, on students from three degree courses in two different Spanish universities. The data therefore attempts to answer a different question. The data presented found significant differences in only 1 group of students which indicates that the methods employed may not be highly effective, yet the conclusions makes a number of unsubstantiated claims about the worthwhile nature of such interventions and therefore their links back to the high level educational initiatives described in some detail at the start. The conclusion is therefore particularly problematic and a description of the 2030 Agenda as part of this section misplaced. A thorough re-write is needed to make clear to the reader what data was collected, and to critically review what it is showing. Showing something is not effective is equally as important as showing that it is, especially if the authors can provide insight into why this was the case.
Author Response

(The authors gave the same response as above.)

Reviewer 3 Report
Dear Authors,
The paper deals with an interesting topic and is well structured, yet I have some comments and suggestions:
- You have stated in line 85-87, that the main objective of the study is „to discover how to develop standards of personal responsibility according to the VBN model in students through service–learning projects”, yet the results of the paper do not underline some standards for developing personal responsibility in students. I am suggesting explaining it better to see a consistency within the paper.
- As your study focuses on the importance of education in raising awareness on matters of responsibility towards the environment and you focused on students, I would suggest including in the theoretical part also the role of the university on this matter.
- You have linked a lot of statements to the VBN model, yet there is little mentioned about this in the paper.
- I suggest including in the sample section also some information about the study year of the students, as you refence it in the results section.
- Figure 2 and Figure 3 still have some remaining words in Spanish that need to be translated.
- I also suggest giving a more representative title to the paper, taking into consideration the information it presents.
Best regards,
Author Response

(The authors gave the same response as above.)
